# Road to Metastasis: The TWEAK Pathway as a Discriminant between Metastasizing and Non-Metastasizing Thick Melanomas

**DOI:** 10.3390/ijms221910568

**Published:** 2021-09-29

**Authors:** Canan Güvenç, Asier Antoranz, Anna Szumera-Ciećkiewicz, Pawel P. Teterycz, Piotr R. Rutkowski, Robert V. Rawson, Richard A. Scolyer, John F. Thompson, Julia Newton-Bishop, Marguerite Stas, Veerle Boecxstaens, Oliver Bechter, Jurgen Vercauteren, Marjan Garmyn, Joost van den Oord, Francesca Maria Bosisio

**Affiliations:** 1Department of Dermatology, University Hospitals Leuven, 3000 Leuven, Belgium; marjan.garmyn@uzleuven.be; 2Department of Imaging and Pathology, Translational Cell and Tissue Research, University Hospitals Leuven, 3000 Leuven, Belgium; asier.antoranzmartinez@kuleuven.be (A.A.); joost.vandenoord@kuleuven.be (J.v.d.O.); 3Department of Pathology and Laboratory Medicine, Maria Sklodowska-Curie National Research Institute of Oncology, 02-781 Warsaw, Poland; szumann@gmail.com; 4Diagnostic Hematology Department, Institute of Hematology and Transfusion Medicine, 02-776 Warsaw, Poland; 5Department of Soft Tissue/Bone Sarcoma and Melanoma, Maria Sklodowska-Curie National Research Institute of Oncology, 02-781 Warsaw, Poland; pawel.teterycz@pib-nio.pl (P.P.T.); piotr.rutkowski@pib-nio.pl (P.R.R.); 6Melanoma Institute Australia, The University of Sydney, Sydney, NSW 2000, Australia; robert.rawson@health.nsw.gov.au (R.V.R.); Richard.Scolyer@health.nsw.gov.au (R.A.S.); John.Thompson@melanoma.org.au (J.F.T.); 7Faculty of Medicine and Health, Sydney Medical School, The University of Sydney, Sydney, NSW 2000, Australia; 8Tissue Pathology and Diagnostic Oncology, Royal Prince Alfred Hospital, Sydney, NSW 2000, Australia; 9NSW Health Pathology, Sydney, NSW 2000, Australia; 10Charles Perkins Centre, The University of Sydney, Sydney, NSW 2000, Australia; 11Department of Melanoma and Surgical Oncology, Royal Prince Alfred Hospital, Sydney, NSW 2000, Australia; 12Institute of Medical Research at St James’s, University of Leeds, Leeds LS2 9JT, UK; J.A.Newton-Bishop@leeds.ac.uk; 13Department of Surgical Oncology, University Hospitals Leuven, 3000 Leuven, Belgium; marguerite.stas@uzleuven.be (M.S.); veerle.boecxstaens@uzleuven.be (V.B.); 14Department of General Medical Oncology, University Hospitals Leuven, 3000 Leuven, Belgium; oliver.bechter@uzleuven.be; 15Department of Microbiology, Immunology and Transplantation, Rega Institute for Medical Research, Clinical and Epidemiological Virology, Katholieke Universiteit Leuven, 3000 Leuven, Belgium; jurgen.vercauteren@kuleuven.be

**Keywords:** melanoma, thick Breslow, TWEAK pathway

## Abstract

Cutaneous melanoma (CM) is the most aggressive form of skin cancer, and its worldwide incidence is rapidly increasing. Early stages can be successfully treated by surgery, but once metastasis has occurred, the prognosis is poor. However, some 5–10% of thick (≥2 mm) melanomas do not follow this scenario and run an unpredictable course. Little is known about the factors that contribute to metastasis in some patient with thick melanomas and the lack thereof in thick melanoma patients who never develop metastatic disease. We were therefore interested to study differential gene expression and pathway analysis and compare non-metastatic and metastatic thick melanomas. We found that the TNF-like weak inducer of apoptosis (TWEAK) pathway was upregulated in thick non-metastasizing melanomas. MAP3K14 (NIK1), BIRC2 (cIAP1), RIPK1, CASP7, CASP8, and TNF play an important role in inhibiting proliferation and invasion of tumor cells via the activation of the non-canonical NF-κB signaling pathway. In particular, this pathway sensitizes melanoma cells to TNF-alpha and activates the apoptosis module of the TWEAK pathway in thick non-metastasizing melanomas. Hence, our study suggests a potential role of the TWEAK pathway in inhibiting thick melanoma from metastasis. Exploitation of these genes and the pathway they control may open future therapeutic avenues.

## 1. Introduction

Cutaneous melanoma (CM) is a tumor that originates from melanocytes of the skin, and continues to carry the potential to be a deadly disease worldwide [1]. While early-stage CM is generally curable as a result of early detection and definitive surgery, diagnosis at a late stage and the consequential delay in treatment has an adverse effect on survival. This is due to the propensity of CM to metastasize, and to the resistance of metastatic cells to classical chemotherapy. Targeted therapies and immunological approaches to treat metastatic disease have dramatically improved survival in patient with CM in recent years [2,3]. Nevertheless, these therapies are not effective in all patients, and are burdened by adverse effects and the development of resistance. It has become crucial to identify biomarkers to assist in identifying patients at highest risk of disease progression to facilitate development of personalized therapy and follow-up care.

The prognosis of primary cutaneous melanoma is correlated with several clinical and histological variables including Breslow-thickness, ulceration, mitotic rate, pattern of tumor-infiltrating lymphocytes, age and gender of the patients and the tumor site. According to Breslow, the thickness of the primary melanoma, measured from the granular layer up to the deepest melanoma cell in the dermis, is by far the most significant prognostic variable and is correlated with outcome [4]. However, some 5–10% of thick classical (≥2 mm) CM do not show this association and have an excellent outcome [5,6,7]. Therefore, investigating the biology of thick melanomas with variable outcome can shed light on important processes driving tumor progression and metastasis. The latter process results from changes in the expression of proteins that favor tumor growth, invasion and migration, such as, for example, the expression of the vasoactive peptide Endothelin-1 or of matrix metalloproteinase 9 (MMP-9) by the CM cells [8]. Several signaling pathways have been identified as key regulators in CM progression, such as the activation of Ras/MAPK signaling pathway through the mutant BRAF or NRAS genes or alterations of CCND1 [9]. Additionally, frequently affected molecular pathways are the PI3K/AKT/mTOR pathways (e.g., PTEN loss of function), cell cycle regulators (e.g., CDKN2a, CDK4, CCND1), P53 (e.g., Tp53, MDM2), and pathways involved in epigenetic regulation (e.g., ARID2a) [10]. Although many studies have investigated the role of distinct features involved in CM spreading, no effort has been made yet to investigate the metastasis of thick CMs starting from a global, unbiased perspective through a bulk sequencing of the transcriptome instead of focusing on the analysis of specific pathways. Moreover, previous studies of gene expression profiles (GEP) in thick CM have not compared groups of patients that have been matched for all other prognostic features that may play a role in influencing the metastatic potential. As a result, the molecular differences between thick non-metastasizing (M−) and thick metastasizing (M+) CMs that may the basis of the different clinical behavior, are not yet fully elucidated. Here, we analyse the significance of classical prognostic parameters in thick CM, and compare the molecular composition of thick primary melanomas that did metastasize with a matched group of thick primary melanomas that did not metastasize, aiming to identify a gene signature associated with non-metastasizing melanomas.

## 2. Results

The analysis of clinicopathological parameters of thick CM are depicted in Appendix A Appendix A (a + b). For the molecular analysis, demographic and clinicopathological characteristics of the study population (thick CM) are depicted in Table 1. The pathway analysis is summarized in Figure 1 and Appendix A Appendix A.

### 2.1. Analysis of Clinicopathological Predictors in Thick CM

The univariate analysis of thick CM (Breslow ≥ 2 mm) from the UZleuven database (*n* = 140), showed no significant association between the development of metastasis within five years and the classical prognostic variables (age, gender, Breslow thickness, ulceration, mitosis, site) plus the additional investigated histopathological variables (subtype, TIL, vascular invasion, solar damage and adjoining nevus). After adjustment for the classical prognostic variables, no significant associations were found between outcome and all investigated parameters in the multivariate analysis (Appendix A Appendix A). We validated our findings in the Leeds database (*n* = 141), where we obtained the same results in the multivariate analysis (Appendix A Appendix A) [14].

### 2.2. Demographic and Clinicopathological Characteristics of the Study Population for the Molecular Analysis

We included 24 patients with thick primary melanoma (≥2 mm) who consented to this study, of whom 6 (25%) were women and 18 (75%) men. We grouped 24 cases into 12 pairs of melanomas that were matched for all clinicopathological prognostic markers but that differed in outcome (i.e., distant metastasis within a follow-up period of 5 years) (Appendix A). Breslow thickness varied from 2.8 mm to 18 mm (median 5 mm, mean 5.7 mm). There were 8 (33.3%) patients younger than 60 years and 16 (66.7%) patients older than 60 years. The primary tumor was located in 11 (45.8%) patients on the extremities, in 6 (25%) on the trunk, and in 7 (29.2%) on the head and neck. Nodular melanoma was the predominant histological subtype of CM in 13 (54.2%) of patients, followed by superficial spreading melanoma (SSM) [4 (16.7%)] and acrolentiginous melanoma (AL) [7 (3.1%)]. Other clinicopathological characteristics are summarized in Table 1.

### 2.3. Molecular Analysis

Of the 24 samples, 23 samples passed the quality control and underwent RNA-sequencing and bioinformatics analysis.

#### 2.3.1. Differentially Expressed Genes in Thick CM

From the 29,457 genes included in the Leuven dataset, we identified 830 genes differentially expressed between M− and M+ (398 overexpressed in M− and 432 overexpressed in M+) However, after correction for multiple testing with the Benjamini–Hochberg method, no significantly differentially expressed genes (DEGs) were found.

#### 2.3.2. Pathway Analysis: TNF-like Weak Inducer of Apoptosis (TWEAK) Pathway

Since complex and heterogeneous phenotypes are best explained by small but coordinated differences in functionally correlated genes, that may therefore not be statistically significant if analyzed separately, we performed pathway analysis. The top ten upregulated pathway in thick M− are depicted in Appendix A Appendix A. We found an upregulation of the TNF-like weak inducer of apoptosis (TWEAK) pathways in non-metastatic thick melanomas. The most affected genes in this pathway were two genes with higher expression in M+ patients (TRAF3 and TRIM63) and 8 genes with higher expression in M− patients (MAP3K14 (NIK1), BIRC2 (cIAP1), RIPK1, CASP7, CASP8, TNF, IL6, MMP9).

#### 2.3.3. Validation in the Leeds Dataset

We validated our findings in the Leeds database. From the 703 primary cutaneous melanomas included in this dataset, 179 met our inclusion criteria (23 M− and 156 M+). From the 20,807 genes, we found 1864 differentially expressed (1016 upregulated in M− and 848 upregulated in M+). No genes remained significant after multiple testing adjustment. The TWEAK pathway in this dataset reported a *p*-value of 9.99 × 10^−5^ (Appendix A Appendix A Volcano plot Leeds database) [14,15].

## 3. Discussion

Despite recent advances in understanding its pathogenesis, melanoma is still the deadliest form of skin cancer. Understanding the molecular mechanisms involved in the development of thick CM, as well as the knowledge of their interplay, has great potential to assist in the clinical management of patients with thick CM. The metastatic process itself in CM is not yet fully understood and gene expression profiling studies for thick primary cutaneous melanomas that did not metastasize are scarce and understudied in the literature. The biology of thick CMs is still largely unexplored and, so far, the underlying mechanisms of metastasis in thick CM have not been investigated with a “multi-omics” approach. In this study we explored the clinical, pathological and molecular differences between non-metastasizing (M−) and metastasizing (M+) thick CMs in order to find elements that could help to explain their different clinical behavior and gain insight into mechanisms of metastasis in CM. Moreover, we tried to find metastatization hallmarks for thick melanomas in order to better stratify patients for prognosis and treatment.

A retrospective analysis of the clinicopathological data of 140 patients with thick CM (≥2.0 mm) from a UZ Leuven cohort with a follow up of at least 5 years revealed that the outcome of these patients was not correlated with any of the known clinical (gender, age, site, etc.) or histological (Breslow thickness, ulceration, mitotic count, TILS solar damage and vascular invasion, nevus) parameters. We also found these observations in the Leeds database. This confirmed that other parameters need to be found to better stratify these selected group of patients (Breslow ≥ 2 mm) than the ones already used in the clinic (according to AJCC8 based on the total group; stage I–IV patients) [4].

These preliminary data encouraged us to enroll patients with thick CM for advanced molecular analysis. In the UZ database and Leeds database, no significantly differentially expressed genes were found, which might be indicative for multifactorial drivers of metastasis, since complex and heterogeneous phenotypes are best explained by small differences in functionally correlated genes. To gain further insight into the molecular changes that occur in thick CM, we also carried out genes set analysis. Pathway enrichment analysis revealed the TWEAK pathway as a potential driving mechanism of metastasis, particularly the submodule that regulates cell death. Hence, our study confirmed for the first time that the apoptosis module of the TWEAK pathway is upregulated in thick M− and identified a gene signature associated with non-metastasizing melanomas. In addition to the TWEAK pathway, the oestrogen receptor pathway was also identified in our gene-set analysis, but we excluded this pathway because hormone-related studies are often multifactorial, and from the literature we could not draw a coherent conclusion on genes that were involved and upregulated in the estrogen receptor pathway in thick CM [16,17].

According to the literature, the TWEAK/Fn14 signaling pathway plays a key role in cancer. While TWEAK and Fn14 gene expression is low in normal healthy tissues, increased expression of these genes has been observed in many solid primary tumor types (kidney, liver, colon, ovarian, esophageal, and pancreatic cancer) [18,19,20,21,22,23,24]. In vivo, TWEAK is a pro-angiogenic [16,17,24] and pro-inflammatory [25,26,27,28] factor that can promote tumor vascularization and inflammation. In addition, studies in vitro have shown that TWEAK-triggered Fn14 activation in cancer cells themselves can stimulate both “protumorigenic/metastatic” and “anti-tumorigenic/metastatic” cellular responses, depending on the investigated cell lines. Armstrong et al. found that TWEAK:Fn14 engagement in cancer cells can positively or negatively regulate the invasive activity, depending on the cancer cell line [18]. They showed that TWEAK activation of the non-canonical NFKB pathway in prostate cancer stimulates proliferation and invasion, while in melanoma cells, it inhibits proliferation and invasion. In particular Fn14-TRAF-TNFR axis regulates intracellular signal transduction that triggers death signals in tumor cells via the non-canonical NF-κB signaling pathway. After the formation of the TRAF2-cIAP1 (BIRC2) complex, it undergoes Cathepsin B mediated degradation. The degradation of the TRAF2-cIAP1 complex leads to the stabilization of MAP3K14 (NIK) [11,12]. The latter induces the autocrine cellular apoptosis machinery by activating the RIPK1-FADDCaspase-8 complex and sensitizes tumor cells to death [11,12]. In our study, we identified six major up-regulated genes involved in the apoptosis module of the TWEAK pathway in thick M− melanoma, namely MAP3K14 (NIK1), BIRC2 (cIAP1), RIPK1, CASP7, CASP8, TNF, suggesting that RIPK1-FADDCaspase-8 complex-mediated apoptosis of melanoma cells may be a crucial mechanism in the prevention of metastasis in thick melanomas. The results presented are in line with a similar study, which also identified members of the NF-κB pathway [14]. Furthermore, we found the strongest log2FC regulation for IL6. IL6 is a classic inducer of STAT3 and induces inflammatory processes. STAT3 was shown to repress MITF and reduces the proliferation of melanoma cells [29].

Our results provide evidence regarding the differential biological behavior of thick melanoma; however, whether our data are also confirmed at the protein level awaits further studies using immunohistochemical techniques on larger numbers of thick melanomas. Our findings highlight the molecular biomarkers that can be used for prognostic purposes and provide information on the underlying mechanisms of outcome in thick CM. Detailed molecular knowledge of thick CM biology will further be instrumental in the specification of new biomarkers for risk stratification, which will improve the clinical management of this disease. In this regard, knowing the apoptotic biomarkers among thick CM patients may be helpful to find better treatment options.

## 4. Materials and Methods

### 4.1. Patient Populations

Patients with high Breslow thickness melanomas with available remaining FFPE tissue were identified from the files of the pathology departments at University Hospitals in Leuven, Belgium (study series; *n* = 4) and at the Melanoma Institute Australia in Sydney, Australia (*n* = 14) and Maria Sklodowska-Curie National Research Institute of Oncology in Warsaw, Poland (*n* = 6). These 24 cases were grouped into 12 pairs of melanomas that were matched for all clinic pathological prognostic markers but that differed in outcome (i.e., locoregional and distant metastasis within a follow-up period of five years). Hence, the study cohort consisted of 12 melanomas that did metastasize and 12 that did not. Each of these 12 pairs were matched for all other prognostic factors as depicted in Appendix A Appendix A. All patients were treated uniformly—i.e., complete excision of the primary melanoma that was followed by re-excision with margins appropriate for the thickness of the primary tumor.

Tissue collections and the specific study protocol were approved by the medical ethical committees and institutional review boards of the University Hospitals at the Katholieke Universiteit Leuven (S61610, approved on 16-08-2018), Melanoma Institute Australia and Maria Sklodowska-Curie National Research Institute of Oncology.

The pathology of each melanoma of our patient cohort was reviewed by four expert dermatopathologists (A. Szumera-Ciećkiewicz, J. J. van den Oord, F.M. Bosisio and R.V. Rawson). Clinical data available included age (<40, 40–59 years, ≥60 years), sex, site of involvement (extremity, head and neck, trunk), histological subtype, mitotic index, ulceration, Breslow thickness and AJCC stage at diagnosis [30]. The clinical and histopathologic characteristics of the study cohort are summarized in Table 1.

### 4.2. Tumor Tissue Samples: RNA Extraction and Purification

Seven consecutive 5 µm sections were prepared from FFPE tissue samples. The first and last sections were stained with hematoxylin & eosin (H & E; Agilent, Santa Clara, CA; USA) and used as a control for the presence of a vertical growth phase and a required tumor cell percentage of at least 10%. After deparaffinization with Histo-Clear and ethanol, tumor foci marked by a pathologist on the H & E section were manually macrodissected. Tissue was digested with proteinase K; RNA extraction was performed using RNeasy FFPE Kit (Qiagen, Hilden, Germany) according to the manufacturer’s instructions, and RNA was quantified using the Qubit 2.0 fluorometer (Thermo Fisher Scientific, Waltham, MA, USA). This study is based on analysis of bulk-RNA, derived from the vertical growth phase of thick CM, and therefore does not take heterogeneity in RNA expression into consideration.

### 4.3. Construction of RNA-Seq Library and Sequencing

Sequence libraries were prepared with the Lexogen QuantSeq 3′ mRNA-Seq Library prep kit according to the manufacturer’s protocol. Samples were indexed to allow for multiplexing. Library quality and size range was assessed using a Bioanalyzer (Agilent Technologies) with the DNA 1000 kit (Agilent Technologies, CA, USA) according to the manufacturer’s recommendations. Libraries were diluted to a final concentration of 2 nM and subsequently sequenced on an Illumina HiSeq4000 platform according to the manufacturer’s recommendations. Single-end reads of 50 bp length were produced with a minimum of 1 M reads per sample.

### 4.4. Bioinformatics Analysis

Quality control of raw reads was performed with FastQC v0.11.7 (Andrews S. (2010)). FastQC: a quality control tool for high throughput sequence data. Available online at: http://www.bioinformatics.babraham.ac.uk/projects/fastqc (accessed on 10 September 2021). Adapters were filtered with ea-utils fastq-mcf v1.05. (Erik Aronesty (2011). ea-utils: “Command-line tools for processing biological sequencing data”; Ea-utils: Fastq processing utilities. https://github.com/ExpressionAnalysis/ea-utils (accessed on 10 September 2021)) Splice-aware alignment was performed with STAR v2.6.1b against the human reference genome hg38 using the default parameters. Reads mapping to multiple loci in the reference genome were discarded [31]. Resulting BAM alignment files were handled with Samtools v1.5 [32].

Quantification of reads per gene was performed with HT-seq Count v2.7.14. Differential gene expression analysis was done comparing thick M- melanomas versus thick M+ melanomas with the DESeq2 [33] R-package (The R Foundation for Statistical Computing, Vienna, Austria). Reported *p*-values were adjusted for multiple comparisons with the Benjamini–Hochberg procedure, which controls false discovery rate (FDR). Pathway analysis was performed using Piano R package for enriched gene set analysis [34]. Within the piano framework, the following gene-set enrichment methods and gene-level statistics were used: Fisher (*p*-value), Stouffer (*p*-value), Reporter (*p*-value), PAGE (t-value), Tail Strength (*p*-value), GSEA (t-value), Mean (logFC), Median (logFC), Sum (logFC), MaxMean (t-value). For the gene-sets we used the Molecular Signatures Database, curated pathways (c2), canonical pathways (cp), version 7.2. Gene-sets with more than 100 genes capturing general biology were removed from downstream analysis (Raw mRNA expression data for UZ Leuven database: primary accession –PRJEB47586 and secondary accession-ERP131867).

### 4.5. Validation Using the Leeds Dataset

The results of our analysis were validated using the Leeds dataset [14]. The normalized expression data and the corresponding patient metadata was obtained directly from the authors. Raw mRNA expression data for Leeds Melanoma Cohort was downloaded from the European Genome Archive using the accession number EGAS00001002922. Thick melanomas were filtered by selecting those patients with a Breslow thickness ≥2 mm. Metastatic patients were defined as those with a time of relapse ≤5 years from the time of diagnosis. Patients with unknown time of relapse were excluded from downstream analysis. Differential gene expression and pathway analysis were performed as described for the Leuven dataset.

## 5. Conclusions

In the present study, we characterized the transcriptome of thick melanoma M− versus M+ in a precisely matched patient data set. We identified a gene signature associated with non-metastasizing melanomas. In particular, the TWEAK-induced activation of the non-canonical NF-κB signaling pathway, which induces production of TNF-alpha and sensitizes tumor cells to death, may play a protective role in thick non-metastasizing melanomas. The exploitation of these genes and pathway may open future therapeutic avenues.

## Figures and Tables

**Figure 1 ijms-22-10568-f001:**
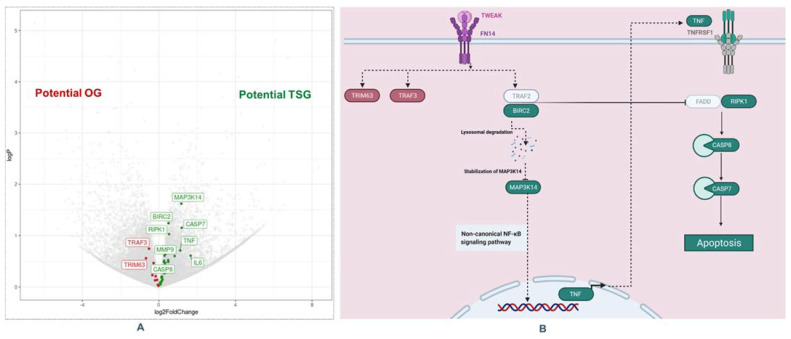
TNF-like weak inducer of apoptosis (TWEAK) Pathway in thick melanoma. (**A**): Volcano Plot for differentially gene expression: red boxes are genes with higher expression in thick M+ (left), and green boxes are genes with higher expression in thick M− (right). (**B**): Apoptosis module of the TWEAK pathway (Created with Biorender): Genes overexpressed in thick M− (green) and genes overexpressed in thick M+ (red), TWEAK/Fn14 interaction may trigger the degradation of cIAPs and then enhances apoptotic processes through TNFR1 [11,12,13]. This figure was adapted from Bhattacharjee et al. Abbreviations: OG: oncogene; TSG: tumor suppressor gene.

**Table 1 ijms-22-10568-t001:** Clinical and histopathological parameters of all patients with thick CM.

Characteristics		Frequency (*n*)	Percent (%)
Metastasis	No	12	50
	Yes	12	50
Subtype	SSM	4	16.7
	NM	13	54.2
	AL	7	29.2
Gender	Female	6	25
	Male	18	75
Ulceration	No	8	33.3
	Yes	16	66.7
Mitosis	No	4	16.7
	Yes	20	83.3
Age	40–59 y	8	33.3
	≥60 y	16	66.7
Site	Extremity	11	45.8
	Trunk	6	25
	Head and neck	7	29.2
Breslow thickness	2–4 mm	12	50
	>4 mm	12	50

Abbreviations: SSM: Superficial spreading melanoma; NM: nodular melanoma; AL: acrolentiginous melanoma. Age is expressed in years and Breslow thickness in mm.

## Data Availability

Raw RNAseq data is available at the European Nucleotide Archive (ENA): primary accession–PRJEB47586, secondary accession-ERP131867.

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
