# Peer review of "Road to Metastasis: The TWEAK Pathway as a Discriminant between Metastasizing and Non-Metastasizing Thick Melanomas"

_ijms, 2021, doi:10.3390/ijms221910568_

Round 1

Reviewer 1 Report

The current study entitled “Road to metastasis: the TWEAK pathway as a discriminant between metastasizing and non-metastasizing thick melanomas” is a very intriguing finding using RNAseq to discover the effect of TWEAK pathway on thickness-induced metastasis. While this finding can be potentially very impactful, this study has some potentially major flaws that must be addressed before recommendation for publication: 

Major comments:

  1. Although the pathway analysis was validated using a Leeds database, there is no experimental validation that this pathway is affected in thick metastasizing or thick non-metastasizing melanomas. Without any type of overexpression or knockdown of the TWEAK pathway, it is very hard to draw definitive conclusions that the pathway has any impact on metastasis.
    1. It would be recommended to knockdown TRAF3 or TRIM63 as well as one of the 8 genes changed in the M- condition and see if metastasis is affected in vitro and/or in vivo.

Minor comments:

  1. The abstract should clearly portray whether TWEAK inhibits or promotes metastasis
  2. In the introduction there was a line that said “Cutaneous Melanoma (CM) is a tumor… carry the potential to be a deadly disease among Caucasians”. This is a misleading line since melanoma is deadly to all races. Melanoma is just most prevalent among Caucasians.

Reviewer 2 Report

The authors submitted a communication article entitled “Road to metastasis: the TWEAK pathway as a discriminant between metastasizing and non-metastasizing thick melanomas”.

When primary cutaneous melanoma is diagnosed, several clinical and histological parameters are collected. Usually it is difficult to assess the individual risk of the patient to develop metastatic lesions. Here, the submitted manuscript deals with the question whether it is possible to detect expressed genes or pathway-signatures, which are associated with metastatic outcome.

The authors found no clinical parameter to be associated with development of metastasis, in two databases. Next, they matched 24 patient-derived samples to form pairs of 2, according to their clinical and histological characteristics. All paraffin sections where macro-dissected and RNA was isolated from tumor foci. Subsquently RNA sequencing was performed and gene expression was analyzed by bioinformatics. Not a single gene with a low FDR-value was identified, but several pathways were found to be regulated significantly. Upregulated pathways in thick, not metastazising melanoma were e.g.: Estrogen receptor pathway (p=0,0009) and TNF related weak inducer of apoptosis TWEAK signaling pathway (p=0,0015).

The results presented are in line with a similar study, which also identified members of the NF-kappaB pathway (Transcriptomic Analysis Reveals Prognostic Molecular Signatures of Stage I Melanoma. Clin Cancer Res. 2019 Dec 15;25(24):7424-7435. doi: 10.1158/1078-0432.CCR-18-3659. Epub 2019 Sep 12. PMID: 31515461.).

Overall, the work presented is important and contributes to unravel mechanisms of melanoma metastasis. The chosen methodology is of very good quality. Analysis and interpretation are up to high standards. Results contribute important findings in the field of melanoma research.

Major points:

  • The authors identified the estrogen pathway with a lower p-value than the TWEAK pathway, why is there no comment on this top ranked pathway? In addition, what pathways are up-regulated in the metastasizing thick melanomas?
  • To verify that the tweak pathway is important, authors could perform a Kaplan Meier analysis. Since metastasis is the reason for patient decease, such an analysis would be suitable.
  • The authors call the genes enriched in non metastatatic, thick melanomas as “potential tumor suppressors”. This could be true, but more likely they just represent the phenotype of the tissue. It is by no means clear that those genes elicit any functional effect. Hence, the terminus tumor suppressor should be avoided.
  • Authors show strongest log2FC regulation for IL6 in FIG1. IL6 is a classic inducer of STAT3 and induces inflammatory processes. STAT3 was shown to repress MITF and lowers proliferation of melanoma cells. This may be worth mentioning.
  • Raw sequencing data should be deposited in public databases to make it available to other researchers.

Minor Points:

  • General:
    • Use “≤”, not “<=” (e.g. 299)
    • Always use the same format to write “supplementary figure/table”, throughout the text large and lower case is mixed up
  • Introduction:
    • 52: not only deadly in Caucasians
    • 67: why inversely? The thicker the worse the prognosis
  • Results:
    • Table 1.: “40-59” -> 40-59y
    • Table 1.: “site” -> Site
    • 126: Use the same format for all titles (up to 2.3. italic, afterwards not)
    • 126: Title numbering wrong, after 2.3 comes 2.2.1
    • 130: only one full stop
    • 135: “pathway” -> pathways
    • 140: Point out “A” and “B” in the figure legend (make bold, add colon character,…)
  • Discussion:
    • 173: “nevus” -> nevus)
    • 173: “leeds” -> Leeds
    • 179: abbreviation already in the results
    • 180: “metastasis.Since” -> metastasis, since
    • 212: “at at” +> at
  • Materials and Methods:
    • 241, 245: full stops are missing
    • 250: why is the reference in brackets?
    • 261: “growht” -> growth
    • 300: why “than”? maybe “from” better?

Round 2

Reviewer 1 Report

Thank you for addressing the comments. The manuscript has been significantly improved as a result of these changes. As a communication this is a very interesting research finding

Reviewer 2 Report

The authors commented and made changes to all review-points raised.

All issues have been addressed.